# Incidence and Risk Factors of Chronic Pulmonary Aspergillosis Development during Long-Term Follow-Up after Lung Cancer Surgery

**DOI:** 10.3390/jof6040271

**Published:** 2020-11-09

**Authors:** Sun Hye Shin, Bo-Guen Kim, Jiyeon Kang, Sang-Won Um, Hojoong Kim, Hong Kwan Kim, Jhingook Kim, Young Mog Shim, Yong Soo Choi, Byeong-Ho Jeong

**Affiliations:** 1Division of Pulmonary and Critical Care Medicine, Department of Medicine, Samsung Medical Center, Sungkyunkwan University School of Medicine, Irwon-ro 81, Gangnam-gu, Seoul 06351, Korea; freshsunhye@gmail.com (S.H.S.); kbg1q2w3e@gmail.com (B.-G.K.); sangwonum@skku.edu (S.-W.U.); hjk3425@skku.edu (H.K.); 2Department of Pulmonology, Inje University Seoul Paik Hospital, Seoul 04551, Korea; qsrsr52@naver.com; 3Samsung Medical Center, Department of Thoracic Surgery, Sungkyunkwan University School of Medicine, Irwon-ro 81, Gangnam-gu, Seoul 06351, Korea; hkts@skku.edu (H.K.K.); jhingookkim@gmail.com (J.K.); ymshim@skku.edu (Y.M.S.)

**Keywords:** chronic pulmonary aspergillosis, lung cancer, surgery

## Abstract

Lung resection surgery for non-small-cell lung cancer (NSCLC) is reportedly a risk factor for developing chronic pulmonary aspergillosis (CPA). However, limited data are available regarding the development of CPA during long-term follow-up after lung cancer surgery. This study aimed to investigate the cumulative incidence and clinical factors associated with CPA development after lung cancer surgery. We retrospectively analyzed 3423 patients with NSCLC who (1) underwent surgical resection and (2) did not have CPA at the time of surgery between January 2010 and December 2013. The diagnosis of CPA was based on clinical symptoms, serological or microbiological evidences, compatible radiological findings, and exclusion of alternative diagnoses. The cumulative incidence of CPA and overall survival (OS) were estimated using the Kaplan–Meier method, and a multivariable Cox proportional hazard analysis was performed to identify factors associated with CPA development. Patients were followed-up for a median of 5.83 years with a 72.3% 5-year OS rate. Fifty-six patients developed CPA at a median of 2.68 years after surgery, with cumulative incidences of 0.4%, 1.1%, 1.6%, and 3.5% at 1, 3, 5, and 10 years, respectively. Lower body mass index (BMI), smoking, underlying interstitial lung disease, thoracotomy, development of postoperative pulmonary complications 30 days after surgery, and treatment with both chemotherapy and radiotherapy were independently associated with CPA development. The cumulative incidence of CPA after surgery was 3.5% at 10 years and showed a steadily increasing trend during long-term follow-up. Therefore, increased awareness regarding CPA development is needed especially in patients with risk factors.

## 1. Introduction

Surgical resection remains the mainstay of curative treatment for non-small-cell lung cancer (NSCLC) [1]. The implementation of low-dose computed tomography (CT) screening has made it possible to diagnose more patients at resectable stages of the disease [2,3]. Additionally, the past few decades saw significant advances in clinical staging and surgical techniques [4,5]. As a result, patients now have a favorable prognosis after surgical resection for NSCLC [6,7].

However, lung cancer surgery is one of the most invasive procedures in medicine and carries considerable risks for developing postoperative pulmonary complications (PPCs) such as pneumonia, empyema, and mediastinitis [8]. While these infectious PPCs mostly occur within 30 or 90 days after surgery [9,10], the impact of the operation may persist beyond the immediate postoperative period and may predispose patients to develop chronic pulmonary infections.

Chronic pulmonary aspergillosis (CPA) is an uncommon but detrimental pulmonary disease caused by members of the genus *Aspergillus* [11] that usually develops in patients with pre-existing pulmonary disease such as tuberculosis (TB) [12,13,14]. A history of treatment for lung cancer or thoracic surgery has been reported as an underlying condition that predisposes patients to CPA [13]. However, there are limited data regarding the development of CPA during long-term follow-up after lung cancer surgery [15].

This study aimed to determine the cumulative incidence of CPA after lung cancer surgery by analyzing a large cohort of patients who complied with long-term follow-up after surgical resection for NSCLC. Furthermore, we also investigated which clinical factors were associated with the development of CPA.

## 2. Methods

### 2.1. Study Population and Data Collection

Patients with NSCLC who underwent surgical resection between January 2010 and December 2014 were identified retrospectively using the Lung Cancer Surgery Registry at Samsung Medical Center, a 1979-bed referral hospital in Seoul, Korea. Patients with a concurrent diagnosis of CPA at the time of surgery were excluded.

The following information was then gathered using the database: patient-related factors such as age, sex, body mass index (BMI), smoking history, underlying pulmonary diseases, and other comorbidities; lung cancer-related factors such as the lobar location of the tumor and the clinical/pathological stage of the disease; and treatment-related factors such as the neoadjuvant treatment used, the extent of surgical resection, the surgical approach, PPCs, and adjuvant treatments administered. Conditions considered as underlying pulmonary disease included a previous history of pulmonary TB, chronic obstructive pulmonary disease (COPD), asthma, or interstitial lung disease (ILD). Tumor stages were determined using the seventh edition of the American Joint Committee on Cancer [16]. Early PPC was defined as the development of pneumonia, acute respiratory distress syndrome or respiratory failure, significant atelectasis requiring bronchoscopy or reintubation, pleural effusion, bronchopleural fistula, empyema, prolonged air leakage lasting for more than 5 days, or pneumothorax during the patient’s hospital stay or during readmission within 30 days after surgery. Late PPC was defined as the development of the same complications more than 30 days after the surgery [17].

This study was approved by the Institutional Review Board of Samsung Medical Center (IRB no. 2020–06-011). The requirement for informed consent was waived owing to its retrospective nature.

### 2.2. Diagnosis of CPA

After surgical resection for NSCLC, most patients were followed-up for at least 5 years by a thoracic surgeon. Patients with pre-existing or newly developed pulmonary disease were jointly followed-up by a pulmonologist. Physical examinations, laboratory tests, chest radiographies, and chest CT scans were regularly performed at scheduled intervals during follow-up visits. When the development of CPA was suspected, patients were referred to a pulmonologist, and further diagnostic tests were performed. The diagnosis of CPA was established on the basis of (1) the presence of compatible clinical symptoms; (2) serological or microbiological evidence of *Aspergillus* infection, including positive serum *Aspergillus* precipitin tests (*Aspergillus fumigatus* Immunoglobulin G Enzyme-linked Immunosorbent Assay kit; IBL International, Hamburg, Germany) and isolation of *Aspergillus* spp. from respiratory specimens, or histological confirmation; (3) radiological findings compatible with evidence of disease progression; and (4) exclusion of alternative diagnoses, according to the widely accepted diagnostic criteria proposed by European Society for Clinical Microbiology and Infectious Diseases/European Respiratory Society [11,18].

### 2.3. Statistical Analyses

Categorical variables are reported as number (%), while continuous variables are reported as median (interquartile range (IQR)). Patients who developed CPA were compared to those who did not develop CPA by using the Pearson χ^2^ test or Fisher’s exact test for categorical variables and the Mann–Whitney test for continuous variables. The Kaplan–Meier method was used to estimate the cumulative incidence of CPA and overall survival (OS) after the surgery. The data of patients who were (1) alive and did not develop CPA, (2) lost to follow-up, and (3) deceased were censored at their last visit to the clinic to analyze the incidence of CPA. For the analysis of OS, the data of patients who were alive or lost to follow-up were censored at the time they were last known to be alive regardless of whether or not they developed CPA.

We used the Cox proportional hazard model to calculate hazard ratios (HRs) and 95% confidence intervals (CIs) for the development of CPA. We entered all relevant variables into the multivariable model and eliminated variables using a stepwise backward selection method (*p* < 0.05 for entry and *p* > 0.10 for removal). Tumor stages were not included in the multivariable model since they had a significant collinearity with neoadjuvant and adjuvant treatments. All tests were two-sided, and a *p* value < 0.05 was considered significant. Predictive Analytics Software (PASW) Statistics 25 (SPSS Inc., Chicago, IL, USA) was used for analysis.

## 3. Results

### 3.1. Study Population and Overall Survival

Out of 3430 patients who underwent surgical resection for NSCLC, seven were diagnosed with CPA by the time of the surgery and excluded from this study (Figure 1). As shown in Table 1, the median age of the study population was 63 years (IQR, 56–69 yeas) and 63.6% were male. The majority of the patients (60.8%) were either current or previous smokers. Video-assisted thoracoscopic surgery (VATS) was performed in 59.4% of the cases (Table 2). Lobectomy was the most common procedure performed (77.0%) followed by sublobar resection (14.4%), bilobectomy (4.5%), and pneumectomy (4.1%).

Patients were followed-up for a median of 5.83 (IQR, 2.92–7.25) years to assess survival, and the 5-year survival rate was 72.3% (Figure 2A).

### 3.2. Development of CPA after Lung Cancer Surgery

During the follow-up period, 56 patients developed CPA at a median of 2.68 (IQR, 1.07–4.53) years after their surgery. The cumulative incidences were 0.4%, 1.1%, 1.6%, and 3.5% at 1, 3, 5, and 10 years, respectively (Figure 2B). Most cases (44/56, 78.6%) involved the ipsilateral side of the resected lung, followed by contralateral (8/56, 14.3%) and bilateral (4/56, 7.1%) development.

Patients who developed CPA were more likely to be male (83.9% vs. 63.3%, *p* = 0.001), smokers (83.9% vs. 60.3%, *p* < 0.001), and have lower BMI (22.1 vs. 23.8 kg/m^2^, *p* < 0.001) than those who did not develop the disease. The presence of pre-existing pulmonary diseases, including history of pulmonary TB, COPD, asthma, and ILD, was not statistically different for the development of CPA. Patients who developed CPA were more likely to have advanced clinical stages of NSCLC (stage III and IV; 50.0% vs. 15.7%, *p* < 0.001) and a histologic diagnosis of squamous cell carcinoma (41.1% vs. 25.0%, *p* = 0.007) (Table 1). Accordingly, a greater percentage of patients who eventually developed CPA received neoadjuvant (37.5% vs. 9.9%, *p* < 0.001) and adjuvant (44.6% vs. 29.8%, *p* = 0.016) therapy than that of patients who did not develop the disease. Regarding surgical procedures, the rate of thoracotomies performed was markedly higher in the group that developed CPA than in the group that did not develop the disease (89.3% vs. 39.8%, *p* < 0.001). Both early (33.9% vs. 18.3%, *p* = 0.003) and late-phase (21.4% vs. 2.4%, *p* < 0.001) PPC were more prevalent in the group that developed CPA (Table 2).

### 3.3. Factors Associated with the Development of CPA

To investigate which factors are independently associated with the development of CPA, both univariable and multivariable analyses were performed (Table 3). Lower BMI (adjusted HR (aHR) per 1 kg/m^2^ increase, 0.83; 95% CI, 0.76–0.91; *p* < 0.001), smoking (aHR, 2.42; 95% CI, 1.16–5.07; *p* = 0.019), underlying ILD (aHR, 5.80; 95% CI, 1.35–25.00; *p* = 0.018), thoracotomy (aHR, 9.60; 95% CI, 3.86–23.85; *p* < 0.001), late-phase PPC (aHR, 6.75; 95% CI, 3.49–13.04; *p* < 0.001), and treatment with both chemotherapy and radiotherapy (aHR, 2.47; 95% CI, 1.33–4.58; *p* = 0.004) were all associated with the development of CPA.

## 4. Discussion

In the present study, patients with NSCLC who underwent surgical resection showed a favorable prognosis with a 5-year survival rate of 72.3%. During long-term follow-up, a small number of patients developed CPA, with a 10-year cumulative incidence of 3.5%. However, the incidence of CPA did not plateau and continued to increase many years following surgery; this suggests that CPA is an important late complication of lung surgery. We also found that factors such as lower BMI, smoking, underlying ILD, and thoracotomy, as well as PPC after 30 days and neoadjuvant/adjuvant therapy with both chemotherapy and radiotherapy, were all independently associated with CPA development.

CPA often develops in patients with pre-existing structural changes of the lung. Pulmonary TB typically leaves cavitation in the lung and has been reported as the most common underlying condition in CPA development; worldwide, an estimated 1.2 million people with CPA have a history of pulmonary TB [12,13]. A recent study has found that CPA complicates 26% of patients who were previously treated for TB and were left with residual pulmonary cavities [14]. Likewise, surgical resection impacts the physical structure of the lung and has been suggested as a risk factor of CPA in previous studies [13,15]. The majority (78.6%) of CPA cases in our study occurred exclusively in the ipsilateral side of the resected lung. Furthermore, patients who underwent open thoracotomy had a risk of developing CPA that was more than nine times higher than that of those who received VATS; this supports the hypothesis that structural sequelae from lung surgery might be the main contributor in the development of CPA. Additionally, late-onset PPC was also associated with the development of CPA during long-term follow-up; this may be due to the development of further pulmonary damage from the complications themselves. Lastly, although it could not reach statistical significance, sublobar resection was associated with a lower risk of CPA, while bilobectomy was associated with a higher risk of CPA development, as compared with lobectomy. This also suggests that dead space formation, prolonged air-leak, or persistent bronchopleural fistula could be the possible mechanism in the pathogenesis of CPA [18,19,20].

Another possible explanation for the development of CPA after lung cancer surgery is the emergence of, to some degree, an impairment of the immune system. In our study, lower BMI, which is a surrogate marker of nutritional status, was an independent risk factor for CPA [21,22]. However, diabetes was not associated with the CPA development in our study [21]. Regarding the use of neoadjuvant and adjuvant therapies, chemotherapy alone was not significantly associated with the development of CPA in the multivariable model. This might be explained by the fact that, unlike in invasive aspergillosis, immunosuppression from cytotoxic agents is not the main pathogenic mechanism in CPA [23]. Rather, treatment with both chemotherapy and radiotherapy increased the risk of CPA, implying that structural damage from radiotherapy plays an important role and may function synergistically with chemotherapy. Similarly, a case report by Hiraki et al. described the development of aspergilloma in a cavity formed from the use of radiofrequency ablation to treat lung cancer [24]. On the other hand, patients who received neoadjuvant or adjuvant treatments have more advanced stages of NSCLC. Thus, these patients were more likely to have undergone additional cancer treatment afterward, which may further increase their vulnerability to these ubiquitous and often opportunistic respiratory pathogens.

Not surprisingly, a considerable number of patients had other underlying pulmonary conditions, aside from lung cancer surgery, such as COPD, asthma, previous pulmonary TB, and ILD. Although COPD and a history of TB were common comorbidities among patients with CPA, there were no significant differences in the prevalence of these conditions between patients with and without CPA. Instead, our study showed that ILD was associated with an increased risk of developing CPA following lung cancer surgery, a finding that is consistent with a previous study [15]. Since the number of patients (*n* = 2) with ILD who also developed CPA was too small, it is difficult to draw a solid conclusion. However, both patients also had idiopathic pulmonary fibrosis (IPF) and neither of them was on steroid or immunosuppressive treatment. Interestingly, both of these patients developed dense consolidations around pre-existing bullae, which corresponds to the idea that expanded air space develops in the residual lung after lobectomy [15,25].

Of note, the 5- and 10-year cumulative incidences of CPA in our study, which were 1.6% and 3.5%, respectively, were lower than those of a previous study from Japan (2.3% and 7.9%, respectively) that was conducted exclusively on lobectomy cases [15]. Since the information regarding the risk factors identified in our study (BMI, surgical approach, PPC, and neoadjuvant and adjuvant therapy use) was not available from the Japanese study, it is difficult to explain what caused this difference. However, there were differences identified regarding the extent of the surgery as our study also included sublobar resection cases. Sublobar resection creates less dead space than lobectomies, which may partly explain the lower incidence of CPA in our study. Another discrepancy from the previous study is the side of the lungs where CPA developed. In our study, 12 (21.4%) patients developed CPA on the contralateral side of the resected lung, whereas in the previous study all cases of CPA affected the same lung that was operated on [15]. Among these 12 patients, 8 had underlying structural abnormalities in the contralateral lung due to sequelae from previous TB (*n* = 3), bronchiectasis (*n* = 1), IPF (*n* = 1), emphysema (*n* = 1), cavitary changes secondary to cancer recurrence (*n* = 1), and subsequent wedge resection on the contralateral lung following cancer recurrence (*n* = 1). Many of these patients also had other risk factors including late-phase PPC (*n* = 3) and use of chemotherapy or chemoradiation therapy (*n* = 3). The remaining four patients who did not have underlying structural abnormalities suffered repeated episodes of pneumonia involving the contralateral lung after the surgery (*n* = 2) or during chemotherapy (*n* = 2). One patient also experienced late-phase PPC. Although the lack of information on the use of neoadjuvant/adjuvant therapies in the previous study prevents further comparison, a considerable number of patients (33.3%) received these treatments in our study, which may have contributed to the development of CPA in the opposite lung.

Our study has several limitations. First, this is a retrospective cohort study from a single referral hospital that handles the largest volume of lung cancer surgeries in the nation. While this may explain the favorable 5-year survival rate in this cohort [26], it may also limit the generalizability of our results. Second, due to its retrospective nature, some patients with CPA might have not been identified since diagnostic tests were performed only when CPA was clinically suspected. Furthermore, the national health insurance system offers special benefits for cancer patients for 5 years starting from the time of diagnosis unless recurrence is documented; this is reflected in the duration of the patients’ follow-up in our study. Indeed, about half (47%) of the cases in the previous study developed more than 5 years after the lobectomy [15]. Thus, the incidence of CPA after lung cancer surgery may have been underestimated in our study. Third, data regarding detailed symptoms and their impact on the patient’s quality of life were not available. Since both the respiratory and systemic symptoms of CPA can negatively impact the quality of life of patients [27], additional comprehensive studies are needed to investigate the effects of CPA on lung cancer survivors.

With the advent of more effective screening methods and improvements in treatment outcomes, more patients with NSCLC are surviving [28]. Among the various problems faced by lung cancer survivors, data regarding CPA remain sparse. Although the incidence of CPA after lung cancer surgery was relatively low at 3.5% at 10 years, our study found a steadily increasing trend during long-term follow-up. Increased awareness regarding this chronic sequela of lung cancer surgery is needed, especially in patients with risk factors.

## Figures and Tables

**Figure 1 jof-06-00271-f001:**
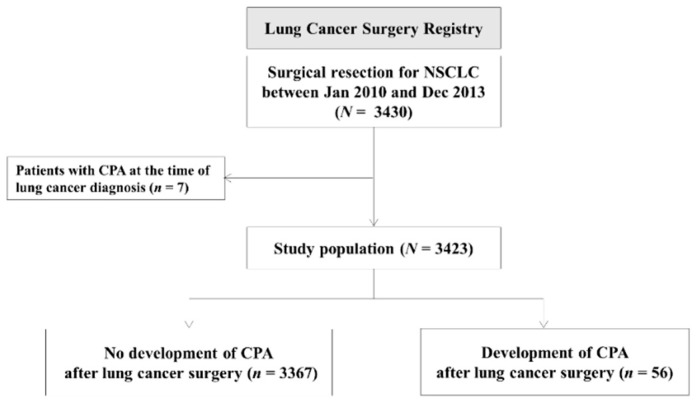
Selection of the study population. NSCLC, non-small-cell lung cancer; CPA, chronic pulmonary aspergillosis.

**Figure 2 jof-06-00271-f002:**
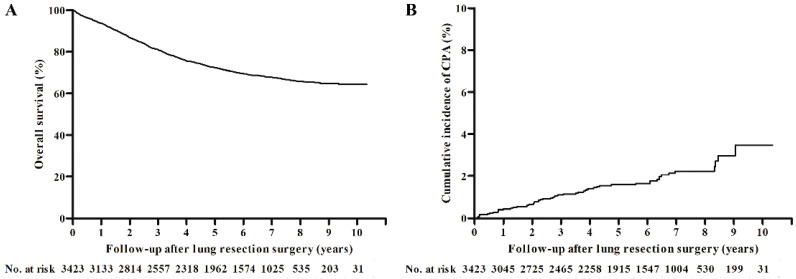
(**A**) Overall survival of the study population and (**B**) cumulative incidence of chronic pulmonary aspergillosis after lung cancer surgery.

**Table 1 jof-06-00271-t001:** Baseline characteristics of patients with NSCLC and the development of chronic pulmonary aspergillosis after lung resection.

Variables	Total (*n* = 3423)	CPA (−) (*n* = 3367)	CPA (+) (*n* = 56)	*p*-Value	CPA Incidence (%)
Age, years	63 (56–69)	63 (56–69)	63 (54–69)	0.789	-
Sex, male	2178 (63.6)	2131 (63.3)	47 (83.9)	0.001	47/2178 (2.2)
Smoking status				<0.001	
Never smoker	1343 (39.2)	1334 (39.6)	9 (16.1)		9/1343 (0.7)
Ex-smoker	1060 (31.0)	1035 (30.7)	25 (44.6)		25/1060 (2.4)
Current smoker	1020 (29.8)	998 (29.6)	22 (39.3)		22/1020 (2.2)
Pack-years (*n* = 2080)	35 (20–50)	35 (20–50)	40 (30–60)	0.059	-
BMI, kg/m^2^	23.8 (21.8–25.7)	23.8 (21.9–25.7)	22.1 (20.5–23.5)	<0.001	-
Comorbidity					
Underlying pulmonary disease					
History of pulmonary TB	385 (11.2)	376 (11.2)	9 (16.1)	0.249	9/385 (2.3)
COPD/Asthma	1158 (33.8)	1140 (33.9)	18 (32.1)	0.788	18/1158 (1.6)
Interstitial lung disease	44 (1.3)	42 (1.2)	2 (3.6)	0.161	2/44 (4.5)
DM	560 (16.4)	550 (16.3)	10 (17.9)	0.760	10/560 (1.8)
Chronic heart disease	213 (6.2)	209 (6.2)	4 (7.1)	0.777	4/213 (1.9)
Chronic renal disease	31 (0.9)	31 (0.9)	0 (0.0)	1.000	0/31 (0.0)
Cerebrovascular disease	144 (4.2)	143 (4.2)	1 (1.8)	0.731	1/144 (0.7)
Previous history of malignancy	479 (14.0)	470 (14.0)	9 (16.1)	0.651	9/479 (1.9)
Clinical stage at diagnosis				<0.001	
Stage I	2254 (65.8)	2236 (66.4)	18 (32.1)		18/2254 (0.8)
Stage II	611 (17.8)	601 (17.8)	10 (17.9)		10/611 (1.6)
Stage III	514 (15.0)	487 (14.5)	27 (48.2)		27/514 (5.3)
Stage IV	44 (1.3)	43 (1.3)	1 (1.8)		1/44 (2.3)
Tumor histology				0.007	
Adenocarcinoma	2317 (67.7)	2290 (68.0)	27 (48.2)		27/2317 (1.2)
Squamous cell carcinoma	866 (25.3)	843 (25.0)	23 (41.1)		23/866 (2.7)
Others^*^	240 (7.0)	234 (6.9)	6 (10.7)		6/240 (2.5)
Location of lung cancer				0.213	
Right	1990 (58.1)	1962 (58.3)	28 (50.0)		28/1990 (1.4)
Left	1433 (41.9)	1405 (41.7)	28 (50.0)		28/1433 (2.0)

Data are presented as *n* (%) or the median (interquartile range). NSCLC, non-small cell lung cancer; CPA, chronic pulmonary aspergillosis; BMI, body mass index; TB, tuberculosis; COPD, chronic obstructive pulmonary disease; DM, diabetes mellitus. *Includes large cell neuroendocrine carcinoma, adenosquamous carcinoma, pleomorphic carcinoma, adenoid cystic carcinoma, mucoepidermoid carcinoma, epithelial myoepithelial carcinoma, and carcinoid tumors.

**Table 2 jof-06-00271-t002:** Treatment profile for NSCLC and the development of chronic pulmonary aspergillosis after lung resection.

Variables	Total (*n* = 3423)	CPA (−) (*n* = 3367)	CPA (+) (*n* = 56)	*p*-Value	CPA Incidence (%)
Neoadjuvant treatment				<0.001	
No	3067 (89.6)	3032 (90.1)	35 (62.5)		35/3067 (1.1)
Yes	356 (10.4)	335 (9.9)	21 (37.5)		21/356 (5.9)
CCRT	299 (8.7)	279 (8.3)	20 (35.7)	<0.001	20/299 (6.7)
Chemotherapy	54 (1.6)	53 (1.6)	1 (1.8)	0.593	1/54 (1.9)
Radiotherapy	3 (0.1)	3 (0.1)	0 (0.0)	1.000	0/3 (0.0)
Surgical approach				<0.001	
VATS	2033 (59.4)	2027 (60.2)	6 (10.7)		6/2033 (0.3)
Thoracotomy	1390 (40.6)	1340 (39.8)	50 (89.3)		50/1390 (3.6)
Types of surgical resection				0.287	
Sublobar resection	492 (14.4)	488 (14.5)	4 (7.1)		4/492 (0.8)
Wedge resection	349 (10.2)	346 (10.3)	3 (5.4)	0.228	3/349 (0.9)
Segmentectomy	143 (4.2)	142 (4.2)	1 (1.8)	0.730	1/143 (0.7)
Lobectomy	2637 (77.0)	2591 (77.0)	46 (82.1)		46/2637 (1.7)
Bilobectomy	153 (4.5)	148 (4.4)	5 (8.9)		5/153 (3.3)
Pneumonectomy	141 (4.1)	140 (4.2)	1 (1.8)		1/141 (0.7)
Pathologic stage (*n* = 3389^*^)				0.006	
I	2094 (61.8)	2073 (62.1)	21 (40.4)		21/2094 (1.0)
II	661 (19.5)	648 (19.4)	13 (25.0)		13/661 (2.0)
III	591 (17.4)	574 (17.2)	17 (32.7)		17/591 (2.9)
IV	43 (1.3)	42 (1.3)	1 (1.9)		1/43 (2.3)
Postoperative pulmonary complication					
Early phase (within 30 days)	636 (18.6)^†^	617 (18.3)	19 (33.9)	0.003	19/636 (3.0)
Late phase (after 30 days)	94 (2.7)^‡^	82 (2.4)	12 (21.4)	<0.001	12/94 (12.8)
Adjuvant treatment (*n* = 3401^§^)				0.016	
No	2380 (70.0)	2349 (70.2)	31 (55.4)		31/2380 (1.3)
Yes	1021 (30.0)	996 (29.8)	25 (44.6)		25/1021 (2.4)
CCRT	191 (5.6)	185 (5.5)	6 (10.7)	0.128	6/191 (3.1)
Chemotherapy	616 (18.1)	608 (18.2)	8 (14.3)	0.466	8/616 (1.3)
Radiotherapy	213 (6.3)	202 (6.0)	11 (19.6)	0.001	11/213 (5.2)

Data are presented as *n* (%). NSCLC, non-small cell lung cancer; CPA, chronic pulmonary aspergillosis; CCRT, concurrent chemoradiotherapy; VATS, video-assisted thoracoscopic surgery. *Except for 34 patients in whom no residual tumors appeared in the surgical specimen after neoadjuvant treatment (pathologic complete response (ypCR)). †Pneumothorax/prolonged air leak (*n* = 312), respiratory failure requiring mechanical ventilation (*n* = 137), pneumonia (*n* = 128), bronchopleural fistula (*n* = 20), others (atelectasis, pleural effusion, etc.) (*n* = 244). Some patients exhibited more than one complication. ‡Pneumonia (*n* = 54), respiratory failure requiring mechanical ventilation (*n* = 25), pneumothorax/prolonged air leak (*n* = 15), bronchopleural fistula (*n* = 15), others (atelectasis, pleural effusion, etc.) (*n* = 31). Some patients exhibited more than one complication. §Excluded 22 patients due to data unavailability.

**Table 3 jof-06-00271-t003:** Prognostic factors associated with the development of chronic pulmonary aspergillosis after lung resection for NSCLC (*n* = 3423).

Variables	Univariable Cox Regression		Multivariable Cox Regression	
	Unadjusted HR (95% CI)	*p*-Value	Adjusted HR (95% CI)	*p*-Value
**Host-related factors**				
Age, years	1.01 (0.98–1.04)	0.524		
Sex, male	3.73 (1.83–7.62)	<0.001		
Body mass index, kg/m^2^	0.82 (0.74–0.90)	<0.001	0.83 (0.76–0.91)	<0.001
Smoking history				
No	Reference		Reference	
Yes	4.25 (2.08–8.67)	<0.001	2.42 (1.16–5.07)	0.019
Comorbidity				
Previous history of pulmonary tuberculosis	1.56 (0.76–3.17)	0.225		
COPD/Asthma	0.95 (0.54–1.66)	0.854		
Interstitial lung disease	5.99 (1.45–24.71)	0.013	5.80 (1.35–25.00)	0.018
Diabetes mellitus	1.30 (0.66–2.57)	0.456		
Previous history of malignancy	1.19 (0.59–2.43)	0.627	1.98 (0.94–4.19)	0.073
**Cancer-related factors**				
Tumor histology				
Adenocarcinoma	Reference			
Squamous cell carcinoma	3.10 (1.77–5.42)	<0.001		
Others^*^	2.95 (1.22–7.15)	0.017		
**Treatment-related factors**				
Surgical approach				
VATS	Reference		Reference	
Thoracotomy	16.69 (7.15–38.96)	<0.001	9.60 (3.86–23.85)	<0.001
Types of Surgical resection				
Lobectomy	Reference			
Sublobar resection	0.45 (0.16–1.24)	0.123		
Bilobectomy	2.22 (0.88–5.60)	0.090		
Pneumonectomy	0.65 (0.09–4.68)	0.664		
Postoperative pulmonary complication				
Early phase (within 30 days) ^†^	2.69 (1.54–4.67)	<0.001		
Late phase (after 30 days) ^‡^	14.60 (7.70–27.69)	<0.001	6.75 (3.49–13.04)	<0.001
Neoadjuvant or Adjuvant treatment				
No	Reference		Reference	
Chemotherapy only	0.84 (0.32–2.20)	0.723	0.43 (0.16–1.14)	0.088
Radiotherapy only	4.31 (1.02–18.25)	0.047	1.47 (0.34–6.37)	0.604
Chemotherapy and radiotherapy both	5.93 (3.38–10.40)	<0.001	2.47 (1.33–4.58)	0.004

NSCLC, non-small cell lung cancer; HR, hazard ratio; CI, confidential interval; BMI, body mass index; COPD, chronic obstructive pulmonary disease; VATS, video-assisted thoracoscopic surgery. *Includes large cell neuroendocrine carcinoma, adenosquamous carcinoma, pleomorphic carcinoma, adenoid cystic carcinoma, mucoepidermoid carcinoma, epithelial myoepithelial carcinoma, and carcinoid tumors. †Pneumothorax/prolonged air leak (*n* = 312), respiratory failure requiring mechanical ventilation (*n* = 137), pneumonia (*n* = 128), bronchopleural fistula (*n* = 20), others (atelectasis, pleural effusion, etc.) (*n* = 244). Some patients exhibited more than one complication. ‡Pneumonia (*n* = 54), respiratory failure requiring mechanical ventilation (*n* = 25), pneumothorax/prolonged air leak (*n* = 15), bronchopleural fistula (*n* = 15), others (atelectasis, pleural effusion, etc.) (*n* = 31). Some patients exhibited more than one complication.

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
