# Peer review of "Incidence and Risk Factors of Chronic Pulmonary Aspergillosis Development during Long-Term Follow-Up after Lung Cancer Surgery"

_jof, 2020, doi:10.3390/jof6040271_

Round 1

Reviewer 1 Report

This is a well performed study/documented survey where the authors explored the risk factors for the development of chronic pulmonary aspergillosis after lung cancer surgery and they followed the patients for at least 10-years after the surgery. They have identified the risk factors to be BMI, smoking, ILD, thoracotomy and PPC.

My only minor comment is that the authors need to have a language check; for example, line 34, correct as ‘fifty-six patients’ and line 101, replace using by ‘use’.

Author Response

This is a well performed study/documented survey where the authors explored the risk factors for the development of chronic pulmonary aspergillosis after lung cancer surgery and they followed the patients for at least 10-years after the surgery. They have identified the risk factors to be BMI, smoking, ILD, thoracotomy and PPC.
C1. My only minor comment is that the authors need to have a language check; for example, line 34, correct as ‘fifty-six patients’ and line 101, replace using by ‘use’.
R1. We corrected line 34 and clarified the line 101.

[Abstract, line 34, page 1]

Fifty-six patients developed CPA at a median of 2.68 years after surgery with cumulative incidences of 0.4%, 1.1%, 1.6%, and 3.5% at 1, 3, 5, and 10 years, respectively.

[Methods, line 101, page 3]

Patients who developed CPA were compared to those who did not develop CPA by using the Pearson χ2 test or Fisher’s exact test for categorical variables and the Mann-Whitney test for continuous variables.

Reviewer 2 Report

This is an interesting retrospective study reporting cumulative incidence of CPA and clinical factors associated with CPA after lung cancer surgery. I think this study will add to the knowledge on epidemiology and clinics of CPA. I have a few suggestions for the authors though:

  1. Modify the title of the study: main aim of the study was to determine the cumulative incidence of CPA post lung cancer surgery. This I think should be part of the title. 
  2. Six instead of sex: In abstract section, line 35, I think authors meant to say fifty-six but it says fifty-sex
  3. I reckon, Cox proportional hazard model is not the ideal analysis to determine clinical risk factors for CPA in this study. Cox model is preferably used to evaluate factors affecting survival in a given condition over certain period of time. To determine clinical factors associated to the develop of CPA in this retrospective cohort, I would highly recommend the authors to carry out a matched nested case-control analysis.    

Author Response

C1. Modify the title of the study: main aim of the study was to determine the cumulative incidence of CPA post lung cancer surgery. This I think should be part of the title.
R1. Thank you for your suggestion. We modified the title to, “Incidence and Risk Factors of Chronic Pulmonary Aspergillosis during Long-term Follow-up after Lung Cancer Surgery”.

C2. Six instead of sex: In abstract section, line 35, I think authors meant to say fifty-six but it says fifty-sex
R2. We corrected this typo.

C3. I reckon, Cox proportional hazard model is not the ideal analysis to determine clinical risk factors for CPA in this study. Cox model is preferably used to evaluate factors affecting survival in a given condition over certain period of time. To determine clinical factors associated to the develop of CPA in this retrospective cohort, I would highly recommend the authors to carry out a matched nested case-control analysis.

R3. We appreciate the reviewer’s comment.

From the beginning of this study, study design and statistical analysis were carried out with support from the statistics experts of “Statistics and Data Center” in our institute. After discussing with statisticians, we implemented the “survival analysis (time-to-event analysis)” since we thought that time from the surgery to CPA development is more important than a mere occurrence of CPA. Thus, as mentioned in the methods section, we followed patients from the time of the lung cancer surgery to the time of CPA development (end point of interest), date of the last follow-up, or the death, whichever comes first. We also aimed to investigate the independent risk factors associated with the CPA development. Our statistician had confirmed that Cox proportional hazards model is appropriate method for this purpose of the study. In addition, colleagues of our department published the study on CPA incidence and risk factors among NTM patients in 2017 [1] (Ref 22 in the manuscript), and Cox regression was also used in that study.  

Of note, we also had inquired whether we should perform competing risk analysis as in the previous study from Japan [2] (Ref 15 in the manuscript). Our statistician recommended using just Cox regression because (1) the incidence of CPA is much lower than the incidence of death, and (2) the actual results of “proportional sub-distribution hazards regression models with death as a competing event” was similar to the Cox regression.

With respect to the reviewer’s comment, we again consulted the statistics team regarding nested case-control analysis and the answers were (1) this is retrospective study with largely variable follow-up durations among patients (and some are censored during follow-up), which makes Cox regression analysis is more appropriate, and (2) the results of nested case-control analysis and Cox regression are usually similar [3]. Therefore, we would like to keep our original analysis using Cox proportional hazard regression. We now have added an acknowledgment to the “Statistics and Data Center” of Samsung Medical Center for their contribution to this study.

Acknowledgment: The authors would like to thank the Statistics and Data Center, Research Institute for Future Medicine of Samsung Medical Center for the statistical analysis. The authors also would like to thank Prof. Jae Il Zo and Prof. Jong Ho Cho of the Department of Thoracic Surgery, and Prof. Kyungjong Lee of the Division of Pulmonary and Critical Care Medicine, Department of Medicine, Samsung Medical Center, Sungkyunkwan University School of Medicine, for providing the research data for this study.

Reference

[1] Tamura, A.; Suzuki, J.; Fukami, T.; Matsui, H.; Akagawa, S.; Ohta, K.; Hebisawa, A.; Takahashi, F. Chronic pulmonary aspergillosis as a sequel to lobectomy for lung cancer. Interact. Cardiovasc. Thorac. Surg. 2015, 21, 650-656.

[2] Jhun, B.W.; Jung, W.J.; Hwang, N.Y.; Park, H.Y.; Jeon, K.; Kang, E.S.; Koh, W.J. Risk factors for the development of chronic pulmonary aspergillosis in patients with nontuberculous mycobacterial lung disease. PLoS One 2017, 12, e0188716.

[3] Essebag, V.; Platt, R.W.; Abrahamowicz, M.; Pilote, L. Comparison of nested case-control and survival analysis methodologies for analysis of time-dependent exposure. BMC medical research methodology 2005, 5, 5.